# Research Progress on Nanotechnology-Driven Enzyme Biosensors for Electrochemical Detection of Biological Pollution and Food Contaminants

**DOI:** 10.3390/foods14071254

**Published:** 2025-04-03

**Authors:** Liang Qu, Xue Zhang, Yanhong Chu, Yuyang Zhang, Zhiyuan Lin, Fanzhuo Kong, Xing Ni, Yani Zhao, Qiongya Lu, Bin Zou

**Affiliations:** 1School of Food and Bioengineering, Wuhu Institute of Technology, Wuhu 241003, China; 101385@whit.edu.cn; 2School of Food and Biological Engineering, Jiangsu University, Zhenjiang 212013, China; 2212318042@stmail.ujs.edu.cn (X.Z.); 19708853623@163.com (Y.C.); 2212218043@stmail.ujs.edu.cn (Y.Z.); 2222218077@stmail.ujs.edu.cn (Z.L.); 2222218033@stmail.ujs.edu.cn (F.K.); 2212318069@stmail.ujs.edu.cn (X.N.); 2212418062@stmail.ujs.edu.cn (Y.Z.); 2222418088@stmail.ujs.edu.cn (Q.L.)

**Keywords:** electrochemical biosensors, nanomaterials, enzyme immobilization techniques, food safety applications

## Abstract

Electrochemical biosensors have attracted widespread attention from researchers due to their simple and rapid operation. Recent advancements in nanobiotechnology have further enhanced their performance, with nanomaterials like graphene, carbon nanotubes, and metal nanoparticles being widely used as carriers for immobilizing enzymes, cells, and DNA molecules. These materials improve stability, sensitivity, and selectivity, making biosensors more effective. This article reviews the introduction, principles, and classification of enzyme-based electrode sensors, as well as their research and application progress in the detection of food risk factors (including foodborne pathogens, biotoxins, drug residues, food additives, allergens, etc.). It also explores future prospects, including advancements in nanotechnology and enzyme immobilization techniques, highlighting their potential in food safety and beyond.

## 1. Introduction

Food safety issues, such as drug residues, contamination by foodborne pathogens, the illegal use of food additives, and biotoxins, have garnered widespread attention [1,2]. Conventional food safety detection methods include liquid chromatography, gas chromatography [3,4], and mass spectrometry [5]; however, these methods suffer from drawbacks such as cumbersome pretreatment processes, high costs, and long detection times. Electrode detection technology is suitable for rapid testing in most industries due to its high automation and quick detection. Nevertheless, the complex environment of food safety testing can cause conventional sensors to be easily interfered with by external factors, resulting in weak detection specificity [6,7]. In 1962, Clark et al. first proposed the concept of combining enzymes with electrodes [8]; five years later, Updike developed an enzyme electrode that was successfully applied for the detection of glucose content in food [9]. Enzyme electrodes have become a focus of attention in current food detection research because of their strong specificity, high sensitivity, and reusability. Zhang et al. [10] prepared a novel glutamate oxidase electrode sensor where the reaction produces hydrogen peroxide, which undergoes electron transfer on the electrode surface. The built-in electronic components convert the electrical signal into a digital signal, allowing for the detection of L-sodium glutamate content in soy sauce. This method does not require soy sauce decolorization or derivatization and can be measured repeatedly over 1000 times. Further studies have demonstrated that high-performance enzyme electrode sensors can enhance and sustain enzyme activity by optimizing the microenvironment on the carrier surface, such as charge distribution and pore structure. The key to this approach lies in the rational design and surface modification of the carrier material. Notably, multi-walled carbon nanotubes (MWCNTs), characterized by their porous hollow structure, high specific surface area, and hydrophobic properties, are extensively utilized in electrochemical sensing and enzyme immobilization applications. Zou et al. [11] developed a high-sensitivity biosensor using multi-walled carbon nanotubes (MWCNTs) functionalized with various groups (-SH, -NH_2_, -Cl, or -OH), alkyl chains, and ionic liquids (-IL1/-IL2) as acetylcholinesterase (AChE) carriers. Figure 1 illustrates the synthesis diagram of the Cl/MWCNTs/AChE/GCE biosensor. Analysis revealed that IL1-modified MWCNTs optimized AChE immobilization, achieving a detection limit of 3.3 × 10^−11^ M for organophosphorus (OP) under optimal conditions (pH 7.0, 0.25 U AChE, 14 min inhibition). The IL1-MWCNTs/AChE/GCE sensor maintained 98.5% reactivity over two weeks and demonstrated 90–104% recovery in vegetable samples, proving its effectiveness for rapid, accurate OP detection. The gradual industrial application of enzyme electrode rapid detection technology has opened up a new chapter in the field of food safety detection.

## 2. Principle of Enzyme Electrode Sensors

The enzyme gel or suspension is immobilized on the surface of the sensitive membrane of the electrode. Compared to free enzymes, immobilized enzymes not only exhibit significantly improved stability but also reduce costs due to their reusable catalytic properties [12,13]. Enzymes are commonly immobilized on electrodes using adsorption, crosslinking, covalent binding, entrapment, layer-by-layer self-assembly techniques, or combinations thereof in order to fully enhance enzymatic catalytic efficiency. Upadhyay et al. [14] prepared a dual-enzyme electrode sensor by immobilizing AChE and choline oxidase using the crosslinking method to detect diisopropyl fluorophosphate and carbofuran in milk, with detection limits of 2.0 × 10^−4^ and 2.0 × 10^−4^ mol/L, respectively. Karra et al. [15] immobilized the enzyme on the electrode using a perfluorosulfonic acid membrane, with the enzyme being covalently linked to carbon nanotubes and chitosan. The detection current of the newly prepared enzyme electrode sensor increased by 1000%. Based on Xia’s research [16], it is known that chloride ionic liquid groups are the optimal carrier for AChE, and functionalizing MWCNTs enhances electron transfer and improves the catalytic environment of AChE. The introduction of horseradish peroxidase (HRP) increases the sensitivity of choline oxidation. The novel dual-enzyme sensor Cl/MWCNTs/HRP/AChE/GCE is used to detect OP residues in vegetables. Figure 2 illustrates the synthesis diagram of the Cl/MWCNTs/HRP/AChE/GCE biosensor. The detection range is 1.0 × 10^−11^–1.0 × 10^−7^ mol/L, with a low detection limit of 4.5 × 10^−12^ mol/L. The reproducibility RSD is 5.6%, the repeatability RSD is 3.0%, the storage stability is above 96.5% within four weeks, and the recovery rate is 83–104%.

The analyte diffuses into the active center of the immobilized enzyme to participate in the catalytic reaction, leading to the generation or reduction of specific active substances. Sensors are classified into amperometric and potentiometric types, both of which measure the content of a specific substance and indirectly reflect the content of another substance [17]. Hatad et al. [18] successfully prepared an amperometric fructose-amino acid oxidase electrode sensor using ruthenium(III) hexammine chloride as the carrier material. Since the response current of the sensor is correlated with the content of glycated albumin, it can measure the content of glycated albumin in actual samples.

The sensor is highly stable and rapid in detection, capable of measuring the content of glycated albumin in protease-digested samples (1.3 μL) within 1 min, with a validity period of up to three months. Potentiometric enzyme electrodes are relatively more complex, and their response current is influenced by environmental temperature, pressure, analyte concentration, and the electrode itself, among other conditions. Hsu et al. [19] prepared a potentiometric sensor based on a urease/bovine serum albumin/Pt carrier material for urea detection. The detected value output between the reference electrode and the working electrode corresponds to the substrate concentration according to the Nernst equation, with a sensitivity of 15.2 mV/dec for urea detection. The sensor exhibits strong specificity for urea detection and has a validity period of 70 days. Amperometric sensors offer a simpler and more intuitive effect compared to potentiometric sensors.

## 3. Classification of Enzyme Electrode Sensors

Enzyme electrodes are typically classified based on the transducer principle and the type of enzymatic catalytic reaction. They are categorized into electrochemical, optical, thermal, piezoelectric, semiconductor, and acoustic enzyme sensors according to different transducer principles; among these categories, research on optical, thermal, and piezoelectric sensors has been limited due to issues with signal stability or the complexity of equipment construction. In contrast, electrochemical enzyme electrode sensors have emerged as the most extensively studied and deeply investigated subject in this field, owing to their superior sensitivity, excellent repeatability, and remarkable selectivity. Shervedani et al. [20] studied a laccase electrode derived from Agaricus bisporus using voltammetry and electrochemical impedance spectroscopy to detect dopamine in plasma and pharmaceutical samples. The results showed that the linear detection ranges were 5.0 × 10^−10^ to 1.3 × 10^−8^ mol/L and 4.7 × 10^−8^ to 4.3 × 10^−7^ mol/L, with correlation coefficients of 0.999 and 0.989, and a detection limit of 2.9 × 10^−8^ mol/L. Ertek et al. [21] made a photoelectric sensor based on the graphite electrode modified by dehydrogenase to detect glucose, with a detection limit of 5.0 × 10^−5^ mol/L; the sensor had excellent sensitivity, repeatability, and selectivity.

Based on the types of enzymatic catalytic reactions, enzyme electrodes can be classified into oxidoreductase electrodes, hydrolase electrodes, isomerase electrodes, transferase electrodes, ligase electrodes, and synthase electrodes. As shown in Table 1, the detection objects, enzyme action mechanism, detection methods, and performance characteristics of enzyme electrodes has been systematically compared according to the types of enzyme-catalyzed reactions. Among these, oxidase/reductase electrodes are the most commonly used. Cui et al. [22] developed a sensor based on multi-walled carbon nanotubes and xanthine oxidase for the rapid and accurate monitoring of intracellular purine content. Under the action of xanthine oxidase, this sensor can provide more information about purine metabolism within cells, opening up a new path for the precise detection of intracellular purines. Kang et al. [23] prepared an enzyme electrode sensor based on acyl-CoA synthetase and oxidase; they found that the electrode detection signal exhibited a good linear relationship with fatty acid concentration, thus enabling the detection of fatty acid concentration.

## 4. Research and Application of Enzyme Electrode Sensors in Food Safety Detection

This article reviews the research progress of enzyme electrode sensors in the safety detection of food and drug residues, biotoxins, food additives, foodborne pathogens, and allergens in recent years [28,29,30,31,32].

### 4.1. Drug Residue Detection

Agricultural product drug residues are classified into veterinary drug residues and pesticide residues. The long-term consumption of meat products with excessive veterinary drug residues can lead to the accumulation of these drugs in the body, causing various acute and chronic poisoning effects. Pesticide residues on the surface of fruits and vegetables often disrupt the balance of hormones in the human body, leading to immune deficiencies and potentially causing cancer, birth defects, and mutations [33,34,35,36,37]. Pesticides are mainly classified into three categories based on their chemical structures: organophosphates, carbamates, and organochlorines.

As organophosphorus pesticides with high toxicity and persistence in agricultural products pose a serious threat to human health and safety, the development of rapid and efficient methods for detecting organophosphorus compounds has gradually attracted attention. Common organophosphorus pesticides on the market include chlorpyrifos, dichlorvos, and trichlorfon, among others. Organophosphorus compounds are a class of acetylcholinesterase inhibitors that readily cause the irreversible phosphorylation of AChE, rendering it inactive. Therefore, it can be inferred that the content of organophosphorus compounds can be indirectly estimated by monitoring the activity of AChE.

Zheng et al. [38] used antibacterial silver nanoparticles and biocompatible chitosan as carrier materials to prepare an AChE electrode for detecting organophosphorus pesticides in agricultural products, with a detection limit for paraoxon of only 1.45 × 10^−5^ mol/L. Miao et al. [39] utilized platinum metal and multi-walled carbon nanotubes as carrier materials to prepare an enzyme electrode for detecting organophosphorus pesticides in agricultural products. The detection limits for malathion, methyl parathion, and chlorpyrifos were 1.6 × 10^−10^, 9.0 × 10^−11^, and 8.0 × 10^−11^ mol/L, respectively. For chlorfenvinphos, the linear range was 5.0 × 10^−8^–5.0 × 10^−7^ mol/L, with a detection limit of 2.97 × 10^−8^ mol/L. This enzyme electrode sensor demonstrated excellent applicability for detecting organophosphorus content in agricultural products.

Itoh et al. [40] prepared a dual-enzyme electrode sensor with AChE and choline oxidase using a gelatin entrapment method. This amperometric enzyme electrode showed a maximum response to acetylcholine chloride at pH 6.8 and 37 °C, with a detection time of only 10 min. The detection limit for trichlorfon was 1 × 10^−10^ mol/L. The enzyme electrode had a shelf life of 30 days, with an average relative deviation of the measured values of 2.18%. Zou et al. [41] combined the advantages of MWCNTs and ionic liquids to construct a dual-enzyme sensor with a Cl/FePP-modified MWCNTs/AChE/glassy carbon electrode. Ferriporphyrin (FePP) was covalently bound via ionic liquids to enhance the sensitivity and stability of the sensor. As shown in Figure 3, MWCNTs are synthesized in situ to obtain Cl-MWCNTs, which are then covalently bound to FePP to generate FePP-MWCNTs and chlorinated ionic liquid. The addition of AChE results in the fabrication of a biomimetic dual-enzyme sensor, where the ionic liquid serves as both an adhesive and an electronic conductor, and FePP catalyzes the oxidation of choline to enhance the sensitivity. When OP inhibits AChE, the inhibition rate can be determined by measuring the current generated during the oxidation process of acetylcholine, enabling the rapid detection of OP residues. The detection limit of this sensor for monocrotophos is 3.2 × 10^−11^ mol/L, with a recovery rate of 89–104%. After 5 weeks of storage, the oxidation current remained at 97.8% of its original value. This biosensor exhibits high stability and sensitivity, making it a promising device for food safety monitoring.

JI et al. [42] developed a multienzyme reaction-mediated electrochemical (MRMEC) biosensor for the sensitive, rapid, and interference-resistant detection of ethephon (ETH). Fe_3_O_4_@Au-Pt and GN-Au NPs were prepared as catalysts and signal amplifiers. Fe_3_O_4_@Au-Pt catalyzes the oxidation of tetramethvlbenzidine (TMB), resulting in a decrease in current. Based on the inhibition of AChE by ETH, differential pulse voltammetry (DPV) was used to detect ETH through the TMB signal, with a limit of detection (LOD) of 2.01 nmol/L. MRMEC effectively analyzed ETH in mangoes, showing satisfactory precision (coefficient of variation: 2.88–15.97%) and recovery rates (92.18–110.72%). As shown in Figure 4, an electrochemical biosensor, which was designed based on nanoenzyme catalysis and cascade enzyme reactions, is used to detect ETH in the aforementioned experiment. This sensor holds promise for detecting various organophosphorus pesticides in food samples.

Carbamate pesticides, first extracted from lentils, are highly toxic and have been extensively synthesized for use on crops due to their effective insecticidal and acaricidal properties to control pest infestations. Currently, major market products include sevin and carbofuran. Zheng et al. [43] prepared an enzyme electrode sensor using ionic liquid-functionalized graphene and gelatin as modifying materials to detect pesticide residues in agricultural products, with a detection limit for sevin of 5.3 × 10^−15^ mol/L.

The sensor retained 95.2% of its initial current after 15 days, demonstrating a strong stability, high sensitivity, and low cost. Carbosulfan, a carbamate pesticide, is widely used in rice production for its ability to control pest growth. Nesakumar et al. [44] employed nano-zinc oxide and platinum-modified AChE enzyme electrodes to measure carbosulfan levels in agricultural products, achieving a detection limit of 2.4 × 10^−10^ mol/L under optimal conditions, with an electrode recovery rate ranging from 99.06% to 100.96%.

Magar et al. [45] designed a selective nanostructured electrochemical biosensing system using six mutants of the EST2 protein from A, with acidocaldarius as the OP-specific biological receptor. The EST2 mutant enzymes were immobilized on disposable screen-printed electrodes modified with Al_2_O_3_/Cu nanocomposites. Comparative studies of the inhibition percentages of the six mutant proteins were conducted over a wide range of pesticide concentrations, achieving a wide dynamic inhibition range with a limit of detection (LOD) of 0.01 nM and a limit of quantification (LOQ) of 0.05 nM for oxon toxicity. The newly developed EST-based biosensor selectively determines paraoxon in different spiked food samples.

Atrazine is a highly effective organochlorine herbicide that inhibits tyrosinase [46]. Based on this principle, Han et al. [47] prepared a tyrosinase electrode sensor using poly(diallyldimethylammonium chloride)-functionalized carbon nanotubes as the carrier material for the detection of atrazine. The detection limit was as low as 1.8 × 10^−8^ mol/L, with a linear range of 4.6 × 10^−8^ to 3.7 × 10^−6^ mol/L. The experiment optimized the inhibition time and the thickness of the electrode’s sensitive surface. This detection method is rapid, convenient, and easy to operate, enabling the efficient detection of atrazine residues in actual samples.

The main detection methods for conventional drug residues include gas chromatography, liquid chromatography, mass spectrometry, and enzyme-linked immunosorbent assay (ELISA). The detection instruments are bulky and complex, and the pre-treatment before detection is cumbersome. Although ELISA is rapid and efficient, it has high detection costs and cannot be used for the real-time, on-site testing of batch products. Enzyme electrode sensors, on the other hand, offer high sensitivity, rapid detection, strong specificity, and require no special pre-treatment of samples, enabling the testing of large batches of samples. In recent years, with continuous exploration by researchers, the performance of enzyme electrodes has gradually been optimized, and there have been numerous research applications for drug residues in agricultural products.

### 4.2. Biotoxin Detection

Toxic substances produced by animals, plants, and microorganisms are referred to as biotoxins. Common biotoxins include the neurotoxin found in pufferfish, gossypol in cottonseeds, and the endotoxin produced by Clostridium botulinum [48]. As biotoxins increasingly threaten human health and safety, enzyme electrode sensors are playing an increasingly important role in biotoxin detection due to their excellent detection sensitivity and accuracy [49,50,51,52,53].

Zhang et al. [54] developed a novel Love Wave sensor-based biosensor using HepG2 cells for the real-time and sensitive detection of okadaic acid (OA), a toxin that causes diarrhetic shellfish poisoning. The sensor unit is detachable and equipped with a miniaturized 8-channel recorder, facilitating experiment preparation and signal measurement. The sensor operates at a synchronous frequency of approximately 160 MHz and is manufactured using ST-cut quartz substrates. HepG2 cells are cultured on the sensor surface as sensing elements in order to record the cell attachment process. The results indicate that the sensor can monitor cell adhesion in real time, with the response signal being correlated to the initial cell seeding density. When the cell-based Love Wave sensor is treated with OA toxin, it exhibits a good performance across various OA concentrations, with a wide linear detection range (10–100 μg/L). Based on the ultrasensitive acoustic wave platform, this biosensor holds promise as a real-time and convenient tool for OA screening.

After a period of time following the death of an organism, highly toxic substances such as histamine, putrescine, and cadaverine, which are produced by microbial decomposition, can pose serious health risks when ingested by humans. Leonardo et al. [55] developed an enzyme electrode sensor based on the combination of diamine oxidase and magnetic beads to detect the levels of histamine, putrescine, and cadaverine, with detection limits within the micromolar range. The sensor exhibits excellent reproducibility (variability below 10%) and repeatability (up to eight consecutive measurements), and has a linear range spanning two orders of magnitude (1.0 × 10^−5^ to 1.0 × 10^−3^ mol/L). Young et al. [56] prepared an amperometric sensor using coenzyme pyrroloquinoline quinone as a modifying material for the detection of histamine. The sensor demonstrates good stability, maintaining no signal loss for up to a month. Under optimal conditions, the detection range is 3.60 × 10^−4^ to 1.53 × 10^−3^ mol/L, with a detection limit of 3.42 × 10^−4^ mol/L. This method is highly practical and can be applied to the detection of food spoilage.

Aflatoxins typically grow in moldy foods with a high starch content, such as nuts and grains. Aflatoxins are highly toxic and carcinogenic. Uludag et al. [57] developed a novel aflatoxin enzyme sensor to detect aflatoxins in wheat spike samples, with a detection limit as low as 6.03 × 10^−9^ mol/L and a recovery rate of 94%. This new biosensor is rapid, sensitive, fully automated, and miniaturized, holding promise for significant economic impact on the food industry in the future. Chen et al. [58] extracted aflatoxin oxidase from Armillaria mellea, which was physically adsorbed onto a gold electrode modified with chitosan and single-walled carbon nanotubes. The prepared enzyme electrode exhibited a significant electrochemical response to sterigmatocystin, with a linear detection range of 10–1480 ng/mL and a detection limit of 3 ng/mL.

Gaelle et al. [59] immobilized a novel recombinant enzyme in polyvinyl alcohol and attached it to a cobalt–phthalocyanine-modified screen-printed electrode, preparing an enzyme electrode for the detection of microcystin content in aquatic products. The results showed a detection limit of 9 × 10^−4^ mol/L; the sensor not only exhibited high sensitivity but also demonstrated superior analytical performance.

β-imidazole naturally occurs in plants (such as *Peganum harmala*), fungi (such as *Claviceps purpurea*), and tobacco, and is an inhibitor of monoamine oxidase. An excessive intake of β-imidazole from different sources may lead to neurological dysfunction and other diseases. Radulescu et al. [60] immobilized the enzyme on a screen-printed electrode using a copper-containing Prussian blue stable film, using benzylamine as the substrate for the enzymatic reaction, developing a rapid analysis method for β-imidazole based on the inhibition of monoamine oxidase. The results showed detection limits of 5.0 × 10^−6^ mol/L for ergotamine and 2.5 × 10^−6^ mol/L for harmaline. This sensor enables the rapid analysis and detection of food. Given the significant harm that toxins pose to humans in various stages of food production, with the continuous optimization of enzyme electrode sensor performance, it is expected to be widely used in the detection of toxins in agricultural products as well.

### 4.3. Food Additive Testing

Food additives are classified into two major categories based on their origin—natural and synthetic. They can improve the color, aroma, taste, preservation, and freshness of food. The excessive or illegal use of food additives can seriously endanger human health and safety. Common food additives include antioxidants, antibiotics, and others [61,62,63,64,65].

Antioxidants are widely used in food, pharmaceutical, and commercial products, and the excessive supplementation of antioxidants can lead to toxic effects. Natural substances, such as catechins, can boost metabolism and slow aging. However, catechins can also be combined with iron, and their excessive intake will reduce the absorption of iron by the body. Sadeghi et al. [66] used Fe_3_O_4_ polyaniline and chitosan as carrier materials to prepare laccase electrode sensors to detect the catechol content in tea. The results showed that the detection range was 5.0 × 10^−7^ to 8.0 × 10^−5^ mol/L and the detection limit was 4.0 × 10^−7^ mol/L. The optimal response time of the enzyme electrode at pH 5, 40 °C was only 8 s. Wang et al. [67] developed a new laccase electrode for the detection of catechol, with a detection limit of 2.9 × 10^−7^ mol/L. The enzyme electrode has good reproducibility, selectivity, and stability. Oil-soluble antioxidants have the ability to cut off the chain reaction of oil oxidation and hinder the continued oxidation of oil; because of their own toxicity, the state stipulates that their use in food should not exceed 0.2 g/kg. Long et al. [68] prepared horseradish peroxidase electrode sensors using gold–platinum nanotubes and graphene as modification materials to detect the content of two antioxidants. The results showed that the detection limits of butylated hydroxyanisole and propyl gallate were 2.55 × 10^−7^ mol/L and 1.13 × 10^−7^ mol/L, respectively. The detection ranges were 1.66 × 10^−5^–2.77 × 10^−4^ mol/L and 4.71 × 10^−7^–4.71 × 10^−4^ mol/L, respectively.

The excessive use of antibiotics can cause a decline in the body’s own resistance. With the increasing demand for quantitative analysis and the detection of antibiotics in agricultural products, researchers continue to explore it. Antibiotics are usually divided into aminoglycosides, tetracycline, macrolides (erythromycin), beta-lactam (penicillin), etc. Taghdisi et al. [69] invented a novel exonuclide I sensor, whose detection limits for tetracycline antibiotics in milk and serum samples were 7.4 × 10^−10^ and 7.1 × 10^−10^ mol/L, respectively. The antibiotic kanamycin belongs to the aminoglycoside family and has a strong antibacterial effect on many enterobacteriaceae bacteria. Song et al. [70] developed a horseradish peroxidase sensor to detect kanamycin in milk, and the results showed that the detection range was 1.72 × 10^−11^–2.57 × 10^−7^ mol/L and the detection limit was 8.58 × 10^−12^ mol/L. The sensor has a strong specificity, sensitivity, and selectivity, and can be applied to the detection of kanamycin in dairy products. Gan et al. [71] used a bare pencil graphite electrode as the working electrode, using imipenem as the substrate, to establish an electrochemical method for measuring the activity of New Delhi metallo-β-lactamase (NDM-1) inhibitors. Upon validation, the IC_50_ values for EDTA and L-captopril were found to be 1.22 ± 0.20 μM and 22.92 ± 1.19 μM, respectively, which are consistent with the results obtained by the UV–visible spectrophotometric method. The steady-state kinetic studies of the typical competitive inhibitors of L-captopril revealed that the K values measured using the two methods were similar, at 41.5 ± 3.8 μM (DPV-PGE) and 52.5 ± 2.5 μM (UV–vis), respectively.

The rational use of synthetic additives has enriched food production and promoted human health, but after all, they are not natural substances and if used improperly or excessively, will seriously harm human health. The enzyme electrode sensor achieves the highly sensitive and rapid detection of food additives and contributes to human health and safety.

### 4.4. Detection of Foodborne Pathogenic Bacteria

Agricultural products are rich in nutrients and are easily contaminated by microorganisms, especially pathogenic microorganisms. The traditional detection methods of food-borne pathogens are time-consuming and complicated, which makes it particularly important to explore a rapid and convenient detection method. Vegetables, meat, dairy products, and seafood are easily contaminated by listeria, which can cause infections of human brain tissue and blood in severe cases [72,73,74,75,76].

Chen et al. [77] invented a urease biosensor for the rapid detection of listeria in milk, with a detection limit of 300 CFU/mL. Salmonella is one of the most serious pathogens that can infect produce. Alkaline phosphatase can catalyze L-ascorbate-2-phosphate to produce L-ascorbic acid, while triphosphine can promote L-ascorbic acid regeneration. Wang et al. [78] designed an alkaline phosphatase electrode sensor based on this REDOX cycle to detect Salmonella. The detection limits of phosphate buffer and agricultural water were 7.6 × 10^2^ CFU/mL and 6.0 × 10^2^ CFU/mL, respectively. Cyanobacteria can produce certain protease inhibitors, causing water pollution. lcer et al. [79] adopted a novel nucleic acid biosensor to achieve the rapid and efficient detection of cyanobacteria in drinking water. The sensor was composed of two sets of gold electrode arrays, including gold nanoparticles modified by antibiotin and protease; the results showed that the detection limit of cyanobacteria was only 6 × 10^−12^ mol/L. Traditional pathogenic bacteria detection is inefficient and slow, such as microbial culture, plate counting, etc. The detection of the enzyme electrode sensor is rapid, efficient, stable, convenient, and low cost; it has great application prospects in the detection of foodborne pathogens.

### 4.5. Food Allergen Detection

Allergens are the antigens that can sensitize the human body, and the body will have an allergic reaction after exposure to these allergens. Allergic reactions usually occur in the body in the form of skin, respiratory, or digestive allergies, and, in severe cases, as systemic allergies and even death. Since the 21st century, there have been more and more cases of allergic diseases, which have become one of the most common global diseases affecting human health [80,81,82,83,84].

Sun et al. [85] invented a new horseradish peroxidase sensor to detect allergens in peanuts, and the results showed that the detection limit was as low as 1.3 × 10^−17^ mol/L. Ruiz et al. [86] invented an electrochemical magnetic bead immune sensor platform for the determination of the allergen β-lactoglobulin in dairy products, horseradine peroxidase-labeled antibody, using antibody modification fixed on the magnetic bead on the surface of the disposable carbon screen-printed electrode; they measured at a potential of −0.20 V, before adding hydroquinone as the electron transfer medium and hydrogen peroxide as the enzyme substrate. With a linear detection range of 2.8–100 ng/mL, the magnetic immunosensor platform has been successfully applied to the detection of β-lactoglobulin in different types of milk.

Amaya et al. [87] determined the content of sensitive gluten in gluten-free foods using electrochemistry, labeled the aptamer on the magnetic bead with streptavitin and peroxidase, and measured the enzyme activity using the time–current method. The activity was inversely proportional to the target concentration in the test solution, and the results showed that when the aptamer had the strongest affinity for the target, the sensor has a detection limit as low as 0.5 mg/L and has an excellent analytical performance.

## 5. Application of Nanomaterials in Enzyme Electrode Sensors

### 5.1. Nanomaterials

Nanomaterials are fine materials whose grain size reaches the nanometer level. When a material reaches the nanometer size, the physical and chemical properties of the material change significantly due to the interaction between the atoms of the material itself. The peculiar effects exhibited by nano-sized substances mainly include interface effects, small-size effects, and macroscopic quantum tunneling effects [88,89,90,91,92].

There are three main stages in the development of nanomaterials. The initial stage is only limited to a single particle powder or single-phase materials. In the second stage, when single-phase nanomaterials gradually cannot meet the demand, researchers began to study a variety of composite nanomaterials. In the third stage, with the increasing development of nanotechnology, researchers began to focus on the study of nano self-assembly and nanoarray systems. Due to their unique physical and chemical properties, nanomaterials have been widely used in the field of electrochemical sensing analysis [29,93,94,95,96]. Based on Zou’s study [11], the functionalization of MWCNTs with ionic liquids after modification with different groups is shown in Figure 5. MWCNTs serve as the carrier material, modified with functional groups including -NH_2_, -OH, -SH, and -Cl, as well as alkyl groups ranging from -CH_3_ to -(CH_2_)_15_CH_3_ and ionic liquids such as Cl^−^ and BF_4_^−^. Subsequently, MWCNTs functionalized with -SH, -(CH_2_)_7_CH_3_, and chloride ionic liquid are used as novel modifiers to improve the performance of AChE.

### 5.2. Nanomaterials Promote the Performance of Enzyme Electrode Sensors

Over the past few decades, nanomaterials have found extensive applications in the field of electrochemical biosensing due to their unique electrochemical properties, significantly enhancing various aspects of the performance of electrochemical biosensors [97,98,99,100,101,102]. Nanomaterials, with their interface effect and small size effect, can significantly improve various performance indicators of biosensors, such as stability, reproducibility, and sensitivity [103]. Moreover, due to the differences in the composition, morphology, and size of various nanomaterials, their roles in the construction process of biosensors are also distinct [104,105,106].

#### 5.2.1. Carbon Nanomaterials

Carbon nanomaterials are derivatives of carbon and are widely used as carrier materials in electrochemical sensing for analytical detection. Examples include carbon nanotubes (CNTs), multi-walled carbon nanotubes (MWCNTs), graphene, mesoporous carbon spheres, crystalline diamond, and diamond-like carbon [107,108,109,110,111,112]. We have described and compared all the carbon nanomaterials mentioned and have presented them in Table 2.

In terms of sensors, compared with the other carbon nanomaterials mentioned in Table 2 above, MWCNTs have the ability to enhance the electrochemical reactivity of important biomolecules and facilitate electron transfer in proteases. Due to the presence of adsorbates, MWCNTs exhibit significant sensitivity to changes in surface conductivity, making them widely used as modification materials in high-sensitivity sensors. Mohammed et al. [116] achieved the precise detection of diclofenac using a novel enzyme electrode based on functionalized MWCNTs and gold–platinum bimetallic nanoparticles (Au-PtNPs). The results showed a detection range of 5 × 10^−10^–1 × 10^−9^ mol/L and a detection limit of 3 × 10^−10^ mol/L. With the assistance of MWCNTs, the sensor exhibited excellent selectivity, strong interference resistance, good reproducibility, repeatability, and stability.

Kaur et al. [117] found that nanocomposites composed of poly3, 4-ethylenedioxyethiophene, and functionalized MWCNTs, as covalently fixed AChE modification materials, could effectively maintain the activity and stability of AChE, and the detection limit of chlorpyrifos, which was prepared based on this biosensor, was as low as 1 × 10^−12^ mol/L. The detection range was 1 × 10^−12^–5 × 10^−8^ mol/L, and the recovery range of realistic samples was 95–97%. Dan et al. [118] found that MWCNTs and gold nanoparticle composites could provide an extremely hydrophilic surface for biomolecular adhesion, and the immobilized AChE showed a good affinity to the substrate; its Km was as low as 2.68 × 10^−11^ mol/L and the detection range for malathion was 3 × 10^−9^–3 × 10^−6^ mol/L. The detection limit was 1.8 × 10^−9^ mol/L.

#### 5.2.2. Other Nanomaterials

In addition to carbon nanomaterials, other nanomaterials are also used in electrochemical biosensors. These materials not only share the common properties of nanomaterials but also exhibit their unique properties due to their different compositions [119,120,121]. For example, nano-metal materials, such as Au, Ag, Pt, etc., have excellent electron transfer ability. Metal oxides, such as Fe_3_O_4_, have magnetic effects, and TiO_2_ has a photoelectric effect. These unique properties will produce unique effects when applied to sensors [122,123,124,125,126]. In addition, nanometallic materials such as Au, Ag, and Pt have excellent electron transfer capabilities. Among metal oxides, Fe_3_O_4_ possesses magnetic effects, while TiO_2_ exhibits photoelectric effects. These unique properties, when applied to sensors, can produce distinct effects [127].

## 6. Conclusions and Outlooks

With the rapid and steady development of agriculture, many problems related to agricultural products have followed, among which pesticide residues are the most concerned. As far as the safety of agricultural products is concerned, the pesticide residues that are more threatening to the human body generally have a huge biological toxicity, such as monocrotophos and methamidophos. AChE is a catalytic enzyme in the animal nervous system, which is involved in the development and maturation of cells and can promote the development of neurons and the regeneration of nerves. When the human body consumes too much OP, it will cause the irreversible phosphorylation of esterase in the body; therefore, the enzyme loses the ability to catalyze the hydrolysis of acetylcholine and gradually ages, so that the transport of the neurotransmitter choline is hindered, thus affecting the function of the central nervous system. By combining enzyme-catalyzed reactions and electrochemical detection principles, such sensors can achieve the highly sensitive and highly selective monitoring of food contaminants, including the real-time detection of pesticide residues, pathogens, food additives, heavy metals, and other harmful substances. Its advantages, such as its convenience, speed, low cost, and strong adaptability, make the enzyme electrode sensor an important technological innovation in the field of food detection.

Although enzyme electrode sensors have been widely used in research on the composition detection of agricultural products, overcoming many limitations of traditional detection methods such as long-time consumption, high costs, and the inability to process in batches, how to rapidly detect specific contaminants using enzyme electrode biosensors has become a major issue in current research in the field of agricultural product safety detection technology. It is also a key factor hindering the industrial application of enzyme electrodes in the agricultural sector. Therefore, it is necessary to further improve the specific recognition of substrates via enzyme electrodes, optimize the stability of enzyme electrodes in use, and overcome the current problem of the rapid inactivation of enzyme electrodes due to environmental influences. In addition, the production process of enzyme electrodes needs to be further simplified to facilitate mechanized production. The poor reproducibility of electrodes during the complex physicochemical preparation process necessitates the further optimization and simplification of the enzyme electrode manufacturing process, while ensuring the sensitivity and service life of the enzyme electrodes.

## Figures and Tables

**Figure 1 foods-14-01254-f001:**
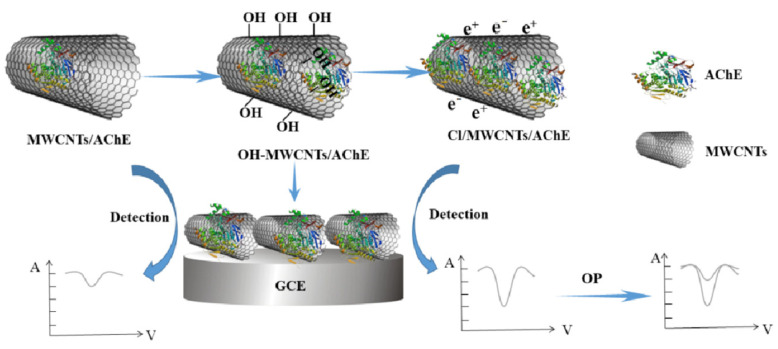
Illustration of the Cl/MWCNTs/AChE/GCE biosensor synthesis test chart (functionalized MWCNTs with diverse groups, alkyl chains, and ionic liquids) [11].

**Figure 2 foods-14-01254-f002:**
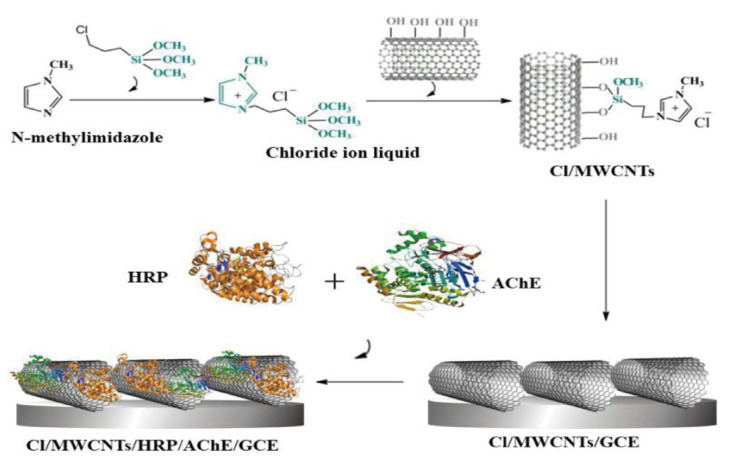
Illustration of the Cl/MWCNTs/HRP/AChE/GCE biosensor synthesis test chart (HRP for choline oxidation enhancement and sensing sensitivity improvement) [16].

**Figure 3 foods-14-01254-f003:**
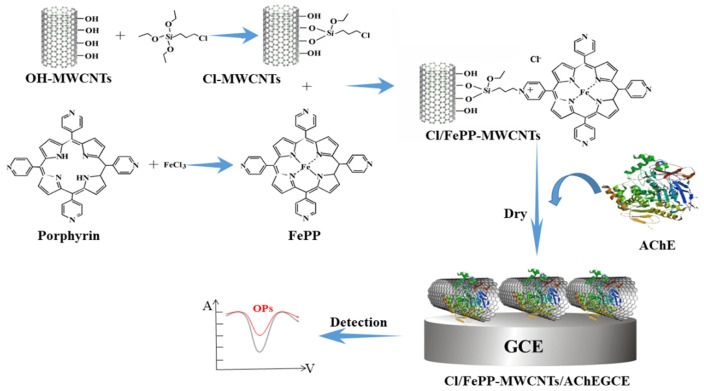
Illustration of the Cl/FePP-MWCNTs/AChE/GCE biosensor synthesis test chart (simulation of oxidase-FePP based on the excellent electrochemical properties of MWCNTs and ionic liquids) [41].

**Figure 4 foods-14-01254-f004:**
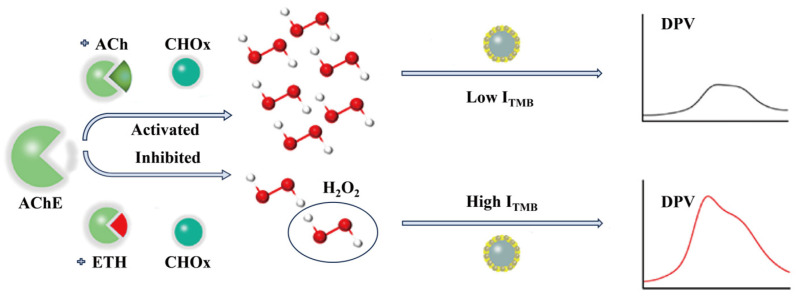
Schematic diagram of enzyme inhibition and the differential pulse voltammetry (DPV) detection of ETH [42].

**Figure 5 foods-14-01254-f005:**
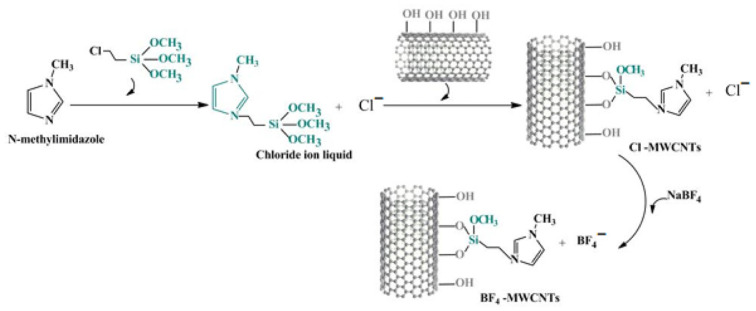
Illustration of Cl/MWCNTs and BF_4_/MWCNTs synthesis (functionalized MWCNTs with -SH, -(CH_2_)_7_CH_3_, and chloride ionic liquids as novel modifiers) [11].

**Table 1 foods-14-01254-t001:** Classification of enzyme electrodes based on enzyme-catalyzed reaction types.

Type	Detection Object	Enzyme Action Mechanism	Method	Performance Feature	Reference
Oxidoreductase electrodes	Intracellular purines	Catalyze the purine oxidation reaction	Electrochemical method	High sensitivity, high accuracy, wide application range, real-time monitoring	[22]
Hydrolase electrodes	Organophosphorus pesticide	Catalyze the hydrolysis of methyl parathion	Electrochemical method	High selectivity, limited sensitivity, limited application range	[24]
Isomerase electrodes	Serum glucose	Catalyze the isomerization of glucose	Electrochemical method	High selectivity, good biocompatibility, limited sensitivity, complex preparation	[25]
Transferase electrodes	Dopamine	Catalyze the isomerization of glucose	Electrochemical method	High selectivity, high accuracy, limited application range	[26]
Ligase electrodes	Ochratoxin A	Catalytic phosphorylation of OTA aptamer connections	Electrochemical method	High selectivity, high sensitivity, complex operation	[27]
Synthase electrodes	Non-esterified fatty acid	Catalyze acyl-coA synthesis reaction	Electrochemical method	High selectivity, high sensitivity, high preparation cost	[23]

**Table 2 foods-14-01254-t002:** Applications of different carbon nanomaterials in electrochemical sensing.

Material Name	Structural Property	The Role of Sensors	Limit of Detection (LOD)	Reference
Carbon nanotubes (CNTs)	Single/multilayer tubular structure, high aspect ratio, excellent electrical conductivity	Enhanced electron transport efficiency, high specific surface area	0.048 ng/mL	[107]
Multi-walled carbon nanotubes (MWCNTs)	Multilayer coaxial tubular structure with defect sites on the surface can be modified functionally	Improve the electrochemical activity of biomolecules, promote electron transfer in proteases, enhanced response to changes in surface conductivity	11.2 pg/mL	[112]
Graphene	Single layer two-dimensional honeycomb structure, high conductivity	Enhance electron transport rate, adsorb target molecules	4.3 nM	[110]
Mesoporous carbon spheres	Porous spherical structure, pore size 2~50 nm, high specific surface area	Provide a large number of enzyme fixation sites, promote molecular diffusion	0.0182 ppb	[113]
Crystalline diamond	Three-dimensional cubic crystal structure, ultra-high hardness and chemical inertness	As electrode base material, provide stable interface	0.07 μM	[114]
Diamond-like carbon	Amorphous carbon material with sp^3^ and sp^2^ hybrid bonds, high wear resistance, and chemical inertness	Electrode surface modification, reduce non-specific adsorption	0.53 μM	[115]

## Data Availability

No new data were created or analyzed in this study. Data sharing is not applicable to this article.

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
