# Peer review of "Research Progress on Nanotechnology-Driven Enzyme Biosensors for Electrochemical Detection of Biological Pollution and Food Contaminants"

_foods, 2025, doi:10.3390/foods14071254_

Round 1

Reviewer 1 Report

Comments and Suggestions for Authors

Research advancements in the utilization of enzyme electrode biosensors for the detection of food risk factor

Thanks for allowing me to evaluate the above said MS.

1. What are the objectives of this study?

2. What are the sources of figures 1-5?

3. Not even a single Tabular data, why?

4. Please don't use the keywords used in the title.

5. In the title authors have used the word food, but in the entire MS, there is no categorization of the food-based detection.

6. Not even a single study belongs to the year 2020-2024, now 2025 already started.

Author Response

Translator        

Reviewer 2 Report

Comments and Suggestions for Authors

It is not understandable why all the Figures have MWCNTs! They are never explained, they are just mentioned and appear!

After reading all the document seems that the authors are only interested in MWCNTs, but this is never explain, neither the review is well focused on this nanomaterial.

Comments on the Quality of English Language

The English is ok. No typo errors were found, and the sentence seems to be well-written. 

Reviewer 3 Report

Comments and Suggestions for Authors

Unfortunately many othet similar paprts are available.It is so difficult to write something of new. In this case more the artuculation of the subject is traditional. So some even important points are omomitted such as pollution of foods from metals, freshness of foods,life cycle of foods.In order to really contribute to sharing of knowledge some new reading key must be found, for instance cost, analytical performance, applied fields (environment, food science, medicine) on choosing case by case the informazionimations deserving to be furnished doing lighter the paper.

Reviewer 4 Report

Comments and Suggestions for Authors

This review article is thought-provoking and articulates significant insights information. However, I recommend that the article be published after some major revisions. Some comments are provided to enhance the quality of the manuscript.

1. The figures lack sufficient detail. Firstly, most of the figures concentrate on MWCNTs, while various other nanoparticles can be associated with enzymes. Secondly, the captions of the figures are not clear enough. Captions should provide more information. It is not indicated whether the figures were sourced from another study. If so, the authors must obtain permission to use the figures due to copyright concerns.

2. Scientific names were inconsistently used in various locations without italicizing, which requires correction.

3. Under sub-section 4.1, Drug residue detection at lines 112 and 113, the topic of Theorem 1 does not make any sense. The authors need to check this part thoroughly.

4. Some abbreviated words were never written anywhere in complete form. For example, ETH, DPV, etc.

5. In sub-section 5.2.1 on carbon nanomaterials, the authors mentioned graphene, mesoporous carbon spheres, crystalline diamond, and diamond-like carbon, in addition to CNTs and MWCNTs. However, they only described the research conducted on MWCNTs. The authors should describe and compare all the carbon nanomaterials they mentioned.

6. Some of the sections lack figures or tables. For example, Sub-sections 5.2.1 and 5.2.1 lack a table. The authors can form a table and compare all the nanomaterials well.

Reviewer 5 Report

Comments and Suggestions for Authors

The abstract is too short, and the novelty cannot be found. What is a new point as compared to other published review articles?

Fig. 1 needs a more general concept that uses enzyme electrode biosensors for multiple targets, as mentioned in the abstract [food risk factors (including 17 foodborne pathogens, biotoxins, drug residues, food additives, allergens, etc.)].

Oervall, the review have discussed about some sensors type, however, it is lacked of comparisons and discussions of these examples.

All Figures are low data and eye-catching. More attractive examples are required to add to the Fig to improve the quality of the manuscript.

Some tables should be added to enhance the quality and information of the reader.

The refs. are only 118, and the manuscript is relatively short as compared to the standard of Foods. Therefore, the authors are strongly commended for adding more information and enriching the references, which give the reader a good comprehensive review.

Round 2

Reviewer 1 Report

Comments and Suggestions for Authors

The authors have updated the MS as per suggestion. 

Author Response

Thank you very much for reviewing our manuscript and providing positive feedback.

Reviewer 3 Report

Comments and Suggestions for Authors

Translator        

I read the new version of the paper with evidence of the variations compared to the first one and I appreciated the efforts of the authors to satisfy my comments.I continue to think that the title is deviating as it does not focus on the novelties of the paper within a field very explored and so difficult to be discovered for really new contributions.These in the paper seem of two natures,nanotechnology applied to food biosensors and particular attention to biological pollution.If these two points could  emerge since ftom  the title the paper should be more appreciated and expecially more read.

Reviewer 4 Report

Comments and Suggestions for Authors

Thank you for carefully revising the manuscript. However, many of the comments are not addressed properly. I am listing those here: 

  1. The captions are not changed, and they do not provide sufficient information. 
  2. It is not indicated whether the figures were sourced from another study. If so, the authors must obtain permission to use the figures due to copyright concerns.
  3. All figures are related to only MWCNTs. No other figures were added that show other nanoparticles. 
  4. In sub-section 5.2.1 on carbon nanomaterials, the authors mentioned graphene, mesoporous carbon spheres, crystalline diamond, and diamond-like carbon, in addition to CNTs and MWCNTs. However, they only described the research conducted on MWCNTs. The author added a table but did not discuss nanomaterials other than MWCNTs. 

Comments on the Quality of English Language

The quality of English can be further improved. 

Reviewer 5 Report

Comments and Suggestions for Authors

The authors have tried to answer all my concerns; however, they are not fully revised examples. Therefore, I would like to ask for further modifications to make significant improvements to the manuscript, which make it meet the requirements to be published in Foods as a comprehensive review article.

Round 3

Reviewer 4 Report

Comments and Suggestions for Authors

The authors made an effort to address all of my comments. Thus, I agree to publish it as is.